# Acceleration of Flower Bud Differentiation of Runner Plants in "Maehyang" Strawberries Using Nutrient Solution Resupply during the Nursery Period

**Hee Sung Hwang** [1], **Hyeon Woo Jeong** [2], **Hye Ri Lee** [2], **Hyeon Gyu Jo** [2], **Hyeon Min Kim** [3] and **Seung Jae Hwang** [1,2,4,5,6,*]

1 Division of Crop Science, Graduate School of Gyeongsang National University, Jinju 52828, Korea; uldangc@naver.com
2 Division of Applied Life Science, Graduate School of Gyeongsang National University, Jinju 52828, Korea; j_dk94@naver.com (H.W.J.); dgpfl77@naver.com (H.R.L.); hk_0871@naver.com (H.G.J.)
3 Division of Plant Resources, Korea National Arboretum, Yangpyeong 12519, Korea; s75364@daum.net
4 Department of Agricultural Plant Science, College of Agriculture & Life Sciences, Gyeongsang National University, Jinju 52828, Korea
5 Institute of Agriculture & Life Science, Gyeongsang National University, Jinju 52828, Korea
6 Research Institute of Life Science, Gyeongsang National University, Jinju 52828, Korea
* Correspondence: hsj@gnu.ac.kr; Tel.: +82-055-772-1916

**Abstract:** The forced cultivation of strawberries (*Fragaria* × *ananassa* Duch.) requires fast flower bud differentiation. Using temporary nutrient-withholding periods is a common management practice for inducing flower bud differentiation at strawberry nurseries in the Republic of Korea. After the temporary nutrient-withholding period, nutrient solution resupply can advance both flower bud growth and fruit yield. This study aims to determine the optimal nutrient solution resupply period with anatomical analysis in order to find a method for fast flower bud differentiation in the early harvest period. Here, the runner plants were divided into 5 groups, each receiving a treatment period of watering (W) and nutrient solution (N) (W40 + N0 (control), W30 + N10, W20 + N20, W10 + N30, and W0 + N40; each number represents the days of the treatment period). The nutrient solution treatments were supplied using a strawberry nutrient solution developed by Gyeongsangnam-do Agricultural Research and Extension. Rapid flower bud differentiation was found for W20 + N20 via anatomical analysis. When the temporary nutrient-withholding period was decreased, the T-N (total nitrogen), P, K, and S concentrations showed a tendency to increase. The C/N ratio showed a tendency to decrease when the nutrient solution resupply period was increased. The W20 + N20 group showed faster flower bud development than the other groups at 10 days before transplanting and on the day of transplanting (2.2 and 5.5), 6 days in a primary cluster budding ratio, and 16 days in flowering plants. No differences in fruit characteristics were observed for the different treatments. In conclusion, the W20 + N20 treatment, which maintains fast flowering, seems to be appropriate for nutrient solution resupply treatment for "Maehyang" strawberries during the nursery period.

**Keywords:** anatomical analysis; budding ratio; C/N ratio; forcing cultivation; fruit yield; total nitrogen

## 1. Introduction

The strawberry (*Fragaria* × *ananassa* Duch.) is a popular fruit crop cultivated in the Republic of Korea. In 2019, the cultivation area of greenhouse strawberries was 6421 ha [1], and, in 2018, the total production of greenhouse strawberries was 181,894 tons [1]. In the past, the most used strawberry cultivation methods have included semi-forced cultivation, which means transplanting runner plants

in October and starting the harvesting the following February. However, now, strawberries are cultivated by forced cultivation, which means transplanting runner plants in September and starting the harvesting in December [2–4]. In the Republic of Korea, forced cultivation is the most used method because of the increased productivity and economic advantages of early harvests [5].

Forced cultivation requires fast flower bud differentiation to bring forward the harvest time, and many researchers have studied methods for earlier flower bud differentiation in strawberries. Providing short daylight conditions and low temperatures using a blackout curtain in the summer season promotes flower bud differentiation in strawberries, based on natural metabolism interaction with the photoperiod and temperature of June-bearing strawberries [6,7]. However, artificially changing the environmental conditions in greenhouses is difficult and increases costs and facility requirements [8]. Another way to promote the flower bud differentiation of strawberries is by changing the nitrogen levels inside the strawberries. When nitrogen levels increase, carbohydrates are used for protein synthesis, and vegetative growth is stimulated [9,10]. Research using root pruning and leaf defoliation to decrease nitrogen levels by decreasing the importation of nitrogen from the roots and old leaves for fast flower bud differentiation has been reported [9,11]. However, root pruning and leaf defoliation cause damage to leaves and roots, which can reduce fruit yields [9,12]. Many researchers have studied lowering strawberry nitrogen levels by regulating the supply of nutrient solutions [13]. When strawberries receive a low level of nutrient solution supply, reproductive growth will be stimulated [14]. However, too much can inhibit flowering and reduce fruit yields [10,15,16]. Consequently, the timing of the temporary nutrient withholding at the nursery stage is important for flower bud differentiation and fruit yields. Many researchers have reported on the timing of temporary nutrient withholding and the effect on flowering and fruit yields in the Republic of Korea [17,18]. Hence, many farmers in the Republic of Korea implement a temporary nutrient-withholding period for their runner plants before transplanting takes place.

However, supplying nitrogen through a nutrient solution after flower bud differentiation advances flower bud development and fruiting [19–21]. Therefore, finding the optimal nutrient solution resupply period is also important. Our previous research showed the possibility of nutrient resupply to promote flower bud differentiation and fruit yields in strawberries [10]. Here, we hypothesize that the optimal nutrient solution resupply after the temporary nutrient-withholding period can promote flower bud differentiation of strawberry plants without sacrificing fruit yield and quality.

In this study, we analyze anatomical images of flower bud differentiation affected by the nutrient solution resupply period and determine the optimal nutrient solution resupply period that promotes flower bud differentiation and enhances the fruit quality and yield of "Maehyang" strawberries, which are the main exportation cultivar when practicing forced cultivation in the Republic of Korea. Additionally, we analyze anatomical data to prove that the nutrient solution resupply period promotes flower bud differentiation at the nursery stage.

## 2. Materials and Methods

### 2.1. Plant Materials and Growth Conditions

Strawberry (*Fragaria* × *ananassa* Duch. cv. Maehyang) runner plants (the plug plant which had not started flower bud differentiation, had 4 to 5 leaves, 9 to 10 mm of crown diameter, transplanted at the same time) were used for this study. On 9 August 2019, the runner plants were transplanted in propagation trays (60 × 34 × 10 cm, 24-cell, Hwaseong Industrial Co., Ltd., Okcheon, Korea) filled with coir (Cocopeat Co., Ltd., Dummalasuriya, Sri Lanka) that had been watered with tap water for 72 h. The tap water analysis showed $Ca^{2+}$ 0.90, $Mg^{2+}$ 0.49, $SO_4^{2-}$ 0.31, and $HCO_3^-$ 0.60 me·L$^{-1}$. $K^+$, $NH_4^+$, $NO_3^-$, and $H_2PO_4^-$ were not detected in the tap water. Nutrient solution treatments of the strawberry plant were supplied using drip tapes and the strawberry nutrient solution developed by Gyeongsangnam-do Agricultural Research and Extension (macroelements: $NO_3^-$ 13.0, $NH_4^+$ 1.0, $H_2PO_4^-$ 4.0, $K^+$ 6.0, $Ca^{2+}$ 8.0, $Mg^{2+}$ 4.0, $SO_4^{2-}$ 4.0 me·L$^{-1}$; microelements: Fe 3.0, B 0.5, Mn 0.5, Zn 0.2,

Cu 0.04, Mo 0.04 mg·L$^{-1}$). During the treatment, the nutrient solution, with an electrical conductivity (EC) of 1.3 dS·m$^{-1}$ and a pH of 5.8, was supplied 3 times a day (5 min with 300 mL per one plant at one time). The tap water was supplied at the same time as the nutrient solution. We controlled the EC and pH levels using a pH/EC meter (HI-98130, Hanna Instruments Co., Ltd., Woonsocket, RI, USA).

After the nutrient solution resupply period, the runner plants were transplanted into pots (295 × 256 × 210 mm, Cheongwoon Industrial Co., Ltd., Gyeongsan, Korea) on 17 September 2019 and hydroponically cultivated until 4 March 2020. During the first week of transplanting, a nutrient solution with EC of 0.6 dS·m$^{-1}$ was provided to induce new root development. Considering the management of appropriate EC levels for strawberry growth stages [8], EC levels were gradually controlled to range from 0.6 to 1.2 dS·m$^{-1}$ (EC 0.6 dS·m$^{-1}$ at the early transplanting stage, EC 0.8 dS·m$^{-1}$ at the budding stage, EC 1.0 dS·m$^{-1}$ at the flowering stage, and EC 1.2 dS·m$^{-1}$ at the harvesting stage). We removed the flowers to control the fruit to 7 fruits on the primary inflorescence and 5 on the secondary inflorescence [4]. The average daytime temperature, nighttime temperature, and relative humidity inside the greenhouse were 22.4 ± 5 °C, 13.0 ± 5 °C, and 39.8 ± 5%, respectively, and these values were measured by a temperature and humidity data logger (TR-74Ui, T&D Co., Ltd., Matsumoto, Japan). Old or excrescent leaves and axillary buds were removed during the cultivation process. Until the flowering stage, germicide and pesticide were sprayed every week to control major diseases and pests, such as powdery mildew, mites, aphids, and *Bradysia agrestis*.

## 2.2. Nutrient Solution Resupply Period

The five treatment groups were set by the different nutrient solution resupply periods (Figure 1). The nutrient solution resupply period was divided into increments of 10 days during the 40-day treatment period. W40 + N0 (control, 40 days of tap water only), W30 + N10 (30 days of tap water and 10 days of nutrient solution), W20 + N20 (20 days of tap water and 20 days of nutrient solution), W10 + N30 (10 days of tap water and 30 days of nutrient solution), and W0 + N40 (40 days of nutrient solution only) were set and started on 9 August 2019 and ended on 17 September 2019.

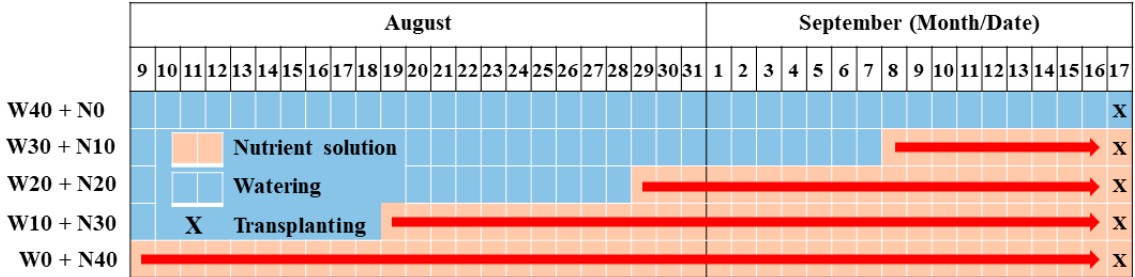

**Figure 1.** Nutrient solution resupply periods of runner plants in "Maehyang" strawberries. W40 + N0, 40 days of tap water only; W30 + N10, 30 days of tap water and 10 days of nutrient solution; W20 + N20, 20 days of tap water and 20 days of nutrient solution; W10 + N30, 10 days of tap water and 30 days of nutrient solution; W0 + N40, 40 days of nutrient solution only.

## 2.3. Flower Bud Differentiation Analysis

We investigated the changes in the flower bud development stage of the growing point to analyze the effects of the nutrient solution resupply period on flower bud differentiation. Twelve plants from each treatment were sampled at 10 days before transplanting and the day of transplanting. All expanded leaves were removed, and unexpanded leaves near the growing point were removed by a scalpel (Feather surgical blade no. 11, FEATHER Co., Ltd., Nakahama, Japan) while looking through a stereoscopic microscope (SMZ 745T, Nikon Co., Ltd., Tokyo, Japan) using the ToupView software package (version 3.7, ToupTek Photonics Co., Ltd., Hangzhou, China). The microscopic view was captured by a microscope digital camera (MSC-M3.0, Mico System Co., Ltd., Seoul, Korea). We referenced the previous results of the report [22] to classify the flower bud development stages into

7 stages. The stages are as follows: 0, the vegetative apex stage; 1, the early apex enlargement stage; 2, the middle apex enlargement stage; 3, the later apex enlargement stage; 4, the apex division stage; 5, the sepal development stage; 6, the stamen development stage.

### 2.4. Analysis of Essential Mineral Element Concentrations

To analyze the essential mineral element concentrations of the runner plants, entire shoots were dried at 70 °C for 72 h using a dry oven (Venticell-222, MMM Medcenter Einrichtungen GmbH., Planegg, Germany) and then crushed into a fine powder using a mortar. One gram of each whole shoot sample was burnt to ashes in a porcelain crucible in a microwave furnace (Model LV 5/11B180, Lilienthal, Berman, Germany) for 4 h at 525 °C. The ash was dissolved in 5 mL of 20% HCl and then in 20 mL of hot distilled water. Next, 25 mL of cold distilled water was added, and the solution was filtered using filter paper. The total nitrogen (T-N), carbon (C), and sulfur (S) were analyzed using a large-capacity automatic element analyzer (TruMAC, LECO, Saint Joseph, MI, USA). Phosphorus (P), potassium (K), calcium (Ca), and magnesium (Mg) were analyzed using the ICP-AES device (OPTIMA-4300DV, PerkinElmer Inc., Waltham, MA, USA).

### 2.5. Growth Characteristics, Flowering Response of Runner Plants

To compare the growth of "Maehyang" strawberry runner plants for the different nutrient solution resupply periods, the petiole lengths, leaf lengths, leaf widths, root lengths, soil plant analysis development (SPAD) values, crown diameters, fresh and dry weights of shoots and roots, and leaf areas after 40 days of treatments were evaluated. Chlorophyll was expressed using the SPAD value, which was measured by a portable chlorophyll meter (SPAD-502, Konica Minolta Inc., Tokyo, Japan). Crown diameters were measured using a Vernier caliper (CD-20PX, Mitutoyo Co., Ltd., Kawasaki, Japan). The fresh weights of shoots and roots were found using an electronic scale (EW220-3NM, Kern & Sohn GmbH., Balingen, Germany) and the dry weights of shoots and roots, which were dried in a dry oven at 70 °C for 72 h, were also measured using an electronic scale. Leaf areas were measured using a leaf area meter (LI-3000, LI-COR Inc., Lincoln, NE, USA). The number of first budding and flowering runner plants from primary and secondary inflorescences was investigated. The budding ratios and flowering ratios of each plant to total runner plants in the primary and secondary inflorescences were calculated.

### 2.6. Fruit Quality and Strawberry Yield

To measure the fruit quality of "Maehyang" strawberries affected by the nutrient solution resupply period, over-90%-mature fruits were sampled at 2- to 3-day intervals for each treatment, and the fruit length, fruit diameter, fruit weight, fruit firmness, soluble solids content, and acidity were measured for each fruit. Fruit firmness was measured with a fruit-specific firmness tester (DFT-01, Proem Co., Ltd., Seoul, Korea), and a 5-mm probe was attached to the same fruit area at a depth of 7 mm. The soluble solids content was measured using a digital refractometer (PR-201a, Atago Co., Ltd., Tokyo, Japan) after removing the 5-mm fruit probe and then measuring the fruit juice, which was then expressed as °Brix. The total acidity was measured with an acidity meter (GMK-835N, GMK Co., Ltd., Seoul, Korea) from 0.3 g of fruit juice in 30 mL of distilled water. The strawberry fruits were sampled from 3 December 2019 to 4 March 2020, and the number of total fruits was observed. Fresh weights over 10 g without a malformed appearance were recorded as the marketable fruit yield.

### 2.7. Statistical Analyses

The experimental treatments were laid out in a completely randomized block design. Each treatment included 72 plants. Twenty-four plants were used per replicate. Experiments were carried out in triplicate. To determine the effect of the nutrient solution resupply period before transplanting, 12 strawberry runner plants per treatment were used to determine all growth characteristics. To analyze the essential mineral elements, 6 runner plant leaves were used for

each treatment. To analyze the flowering response and total fruit yield and quality of strawberries after transplanting, 12 plants were used for each treatment. The statistical analyses were carried out using the Statistical Analysis System program (SAS 9.4, SAS Institute Inc., Cary, NC, USA). The experimental results were subjected to an analysis of variance (ANOVA) and Tukey's test. Significant differences were considered at $p \leq 0.05$. The graph was plotted using the SigmaPlot software package (SigmaPlot 12.5, Systat Software Inc., San Jose, CA, USA).

## 3. Results and Discussion

### 3.1. Effect of Nutrient Solution Resupply Period on Flower Bud Differentiation

As shown in Table 1, 10 days before transplanting, flower bud differentiation showed no significant differences between all treatments except for W20 + N20, where the group featured more developed flower buds at Stage 2.2 compared to the other treatments. On the day of transplanting, W20 + N20 and W10 + N30 had significantly high developed flower bud stages at 5.5 and 5.1, respectively (Table 1 and Figure 2). Additionally, W40 + N0 had a significantly low developed flower bud stage at 2.4. When flower bud differentiation starts, the nutrient solution supply can help the development of flower buds [20,21]. It has been reported that additional nitrogen can bring forward the date of strawberry (cv. Ai-Berry) flower bud development by about 7 days [23]. However, if strawberries receive continuously high nutrition, this can delay flower bud differentiation [24,25]. The reason for this is the distribution of assimilation products, where too much nitrogen promotes vegetative growth, caused by most assimilation products being used for protein synthesis, and too little nitrogen induces poor growth in plants [10]. When nutrient supply was significantly reduced, the flowering rate did not increase [26,27]. Consequently, the W20 + N20 treatment was considered advantageous for the flower bud development of "Maehyang" strawberries.

**Table 1.** Average flower bud development stages on the apical meristem of "Maehyang" strawberry runner plants affected by the nutrient solution resupply period at 10 days before transplanting and on the day of transplanting (*n* = 12).

| Treatment [z] | Average Flower Bud Development Stage [y] | |
|---|---|---|
| | 10 Days before Transplanting | The Day of Transplanting |
| W40 + N0 | 0.7 b [x] | 2.4 c |
| W30 + N10 | 1.0 b | 4.3 ab |
| W20 + N20 | 2.2 a | 5.5 a |
| W10 + N30 | 1.0 b | 5.1 a |
| W0 + N40 | 1.1 b | 3.3 bc |

[z] Refer to Figure 1 for details on nutrient solution resupply periods. [y] The flower bud development stage was classified into 7 stages (0 to 6). The stages are as follows: 0, the vegetative apex stage; 1, the early apex enlargement stage; 2, the middle apex enlargement stage; 3, the late apex enlargement stage; 4, the apex division stage; 5, the sepal development stage; 6, the stamen development stage. [x] Mean separation within columns by Tukey's test at $p \leq 0.05$.

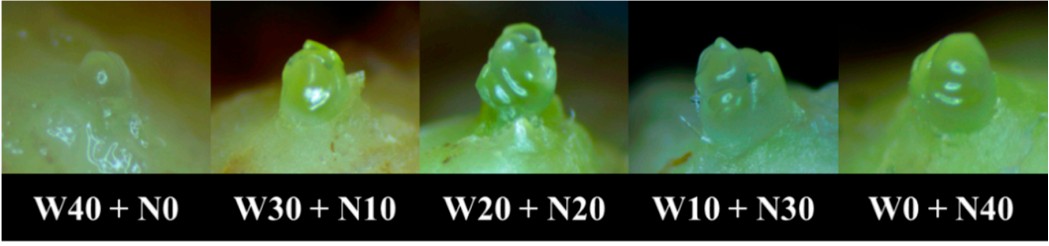

**Figure 2.** Microscopic view of flower bud development stages on the apical meristem of "Maehyang" strawberry runner plants affected by the nutrient solution resupply period on the day of transplanting. Refer to Figure S1 for details on images of flower bud development stages. Refer to Figure 1 for details on nutrient solution resupply periods.

### 3.2. Essential Mineral Element Concentrations of Runner Plants Shoots

Table 2 shows the concentrations of macroelements in the shoots of "Maehyang" strawberry runner plants. When the nutrient solution supply period was increased, the T-N, P, K, and S concentrations of runner plants significantly increased. The W0 + N40 treatment showed the highest value of all the treatments. When a nutrient solution is supplied to a plant for a long period, it tends to absorb more elements and accumulate them [28]. The concentration of C was high in W20 + N20, W10 + N30, and W0 + N40. When plants enter the flower bud differentiation stage, carbon content is increased for carbohydrate accumulation and the preparation of ethylene production [29,30]. The C/N ratio showed a significant decrease when the nutrient solution resupply period was increased. The W0 + N40 treatment showed the lowest value of all the treatments. This was thought to be due to an increase in the T-N concentration. T-N and P play important roles in modulating vegetative and reproductive growth [13], and an excessively high concentration of nitrogen in plants can inhibit flower bud differentiation [21]. Therefore, a high C/N ratio induces high flower bud differentiation [31]. In addition, a P deficiency decreases flower development [32]. It was reported that the Ca concentration increases when strawberries flower because of cell division and the cytokinin active process [33]. The Ca concentration was high in the W30 + N10, W20 + N20, and W10 + N30 groups, with 1.89–2.03%. The Mg concentration showed no significant differences between all treatments.

**Table 2.** Concentration of macroelements in the shoots of "Maehyang" strawberry runner plants affected by the nutrient solution resupply period during the nursery stage ($n = 6$).

| Treatment [z] | T-N [y] | C | P | K | Ca | Mg | S | C/N Ratio [x] |
|---|---|---|---|---|---|---|---|---|
| | | | | (%) | | | | |
| W40 + N0 | 1.18 d [w] | 35.71 b | 0.20 c | 0.98 c | 1.71 bc | 0.44 a | 0.09 b | 30.55 a |
| W30 + N10 | 1.91 c | 35.61 b | 0.51 b | 1.57 b | 1.94 ab | 0.48 a | 0.11 b | 18.69 b |
| W20 + N20 | 2.19 b | 37.14 a | 0.61 ab | 1.57 b | 2.03 a | 0.49 a | 0.10 b | 17.02 bc |
| W10 + N30 | 2.18 b | 35.84 ab | 0.67 a | 1.81 ab | 1.89 ab | 0.50 a | 0.11 b | 16.54 bc |
| W0 + N40 | 2.46 a | 35.77 ab | 0.68 a | 1.98 a | 1.64 c | 0.48 a | 0.12 a | 14.58 c |

[z] Refer to Figure 1 for details on nutrient solution resupply periods. [y] Concentration of total nitrogen. [x] C/N ratio is the value of the carbon divided by total nitrogen. [w] Mean separation within columns by Tukey's test at $p \leq 0.05$.

### 3.3. Growth Characteristics of Runner Plants

Tables 3 and 4 summarize the growth characteristics of runner plants for the different nutrient solution resupply periods. The petiole lengths, leaf widths, and fresh weights of shoots were the highest in the W10 + N30 group, and leaf lengths were the highest (at 10.76 cm) in the W0 + N40 group. Nitrogen is a major element for the control of growth and plant architecture in the nutrient solution [13]. When the nutrient solution was supplied at the strawberry nursery stage, this was shown to increase plant height, fresh weight, and the number of new leaves as compared to only the supply of tap water [17,34,35]. The SPAD value is usually used as an index to represent the chlorophyll content [36]. The SPAD value was lowest (at 42.93) in the W40 + N0 group. The high nitrogen level inside plants induces vegetative growth and an increase in the chlorophyll content [17,37]. The T-N level inside the W40 + N0 group was the lowest when compared to the other treatments (Table 2). Low nutrient conditions cause a decrease in photosynthesis, metabolization, and hormone signaling, which results in a decrease in growth [38]. The fresh and dry weights of roots showed a significant decrease when increasing the nutrient solution resupply period. The W0 + N40 treatment was the lowest in the fresh weights of roots. The W10 + N30 treatment was the lowest in the dry weights of roots.

This is similar to previous studies when plants were moved from high-nutrient to low-nutrient conditions; root growth was increased [17,39]. This phenomenon has been explained as an adaptation to foraging for nutrients in a low-nutrient environment [40]. Additionally, the excessive growth of shoots can inhibit the growth of roots [41]. However, the root lengths, crown diameters, dry weights of shoots, and leaf areas were not significantly different between all treatments. The crown diameters, shoot dry

weights, and leaf areas were highly correlated with fruit yield in strawberries, and these properties are used as the main indicators of runner plant quality [42–44]. In this study, some morphological changes were found, such as the petiole length, leaf length, and leaf width, but the main indicators of runner plant quality (crown diameter, shoot dry weight, and leaf area) showed no significant differences. Accordingly, it was considered that the quality of the runner plants was not affected by the nutrient solution resupply period.

**Table 3.** Growth characteristics of "Maehyang" strawberry runner plants affected by the nutrient solution resupply period during the nursery stage (*n* = 12).

| Treatment [z] | Petiole Length (cm) | Leaf Length (cm) | Leaf Width (cm) | Root Length (cm) | SPAD Value | Crown Diameter (mm) |
|---|---|---|---|---|---|---|
| W40 + N0 | 27.79 b [y] | 8.93 c | 6.03 b | 22.60 a | 42.93 b | 10.20 a |
| W30 + N10 | 25.93 b | 9.37 bc | 6.26 ab | 18.95 a | 46.28 a | 10.21 a |
| W20 + N20 | 29.03 ab | 9.47 bc | 6.30 ab | 20.52 a | 46.30 a | 10.37 a |
| W10 + N30 | 31.53 a | 10.28 ab | 6.90 a | 20.61 a | 45.38 a | 9.85 a |
| W0 + N40 | 29.12 ab | 10.76 a | 6.65 ab | 21.72 a | 46.30 a | 10.28 a |

[z] Refer to Figure 1 for details on nutrient solution resupply periods. [y] Mean separation within columns by Tukey's test at $p \leq 0.05$.

**Table 4.** The fresh and dry weights and leaf area of "Maehyang" strawberry runner plants affected by the nutrient solution resupply period during the nursery stage (*n* = 12).

| Treatment [z] | Fresh Weight (g/plant) | | Dry Weight (g/plant) | | Leaf Area (cm$^2$/plant) |
|---|---|---|---|---|---|
| | Shoot | Root | Shoot | Root | |
| W40 + N0 | 18.34 b [y] | 4.60 a | 4.61 a | 0.76 ab | 417.53 a |
| W30 + N10 | 17.51 b | 4.19 ab | 4.14 a | 0.78 a | 419.63 a |
| W20 + N20 | 21.47 ab | 4.03 ab | 4.24 a | 0.61 a–c | 501.40 a |
| W10 + N30 | 23.93 a | 3.50 ab | 4.58 a | 0.52 c | 544.33 a |
| W0 + N40 | 20.33 ab | 3.22 b | 3.98 a | 0.55 bc | 475.13 a |

[z] Refer to Figure 1 for details on nutrient solution resupply periods. [y] Mean separation within columns by Tukey's test at $p \leq 0.05$.

### 3.4. Flowering Response of Runner Plants

The budding ratio and flowering plants of the primary and secondary inflorescences are shown in Figure 3. The primary budding occurred first in the W20 + N20 group before 22 October 2019. The W20 + N20 and W30 + N10 groups achieved the first 100% primary budding rate on 30 October 2019 (Figure 3A). The first primary flowering plants belonged to the W20 + N20, and W30 + N10 groups, and this occurred on 4 November 2019. Additionally, the W20 + N20 group featured the first 100% primary flowering plants, which occurred on 28 November 2019 (Figure 3B). The W20 + N20 group featured the most developed flower bud just before the day of transplanting (Table 1). Unlike the W40 + N0 (which had a high C/N ratio but was low in other essential minerals) and W0 + N40 (which had a high nitrogen content and a low C/N ratio) groups, the W20 + N20 group was positioned in the middle of the range of element contents (Table 2). The secondary budding ratio occurred first in the W20 + N20 group before 19 November 2019. However, the W30 + N10 group achieved the first 100% secondary budding ratio on 3 December 2019 (Figure 3C). Secondary flowering plants were found first in the W20 + N20 group before 1 December 2019. However, the W30 + N10 group found the first 100% secondary budding ratio on 18 January 2020 (Figure 3D). For the secondary inflorescences, the effect of the nutrient solution resupply period seems to be relatively low; however, some nutrient solution resupply period groups (W30 + N10, W20 + N20) had a better budding ratio and flowering of plants than the W40 + N0 and W0 + N40 groups. It is known that fast flowering induces an early harvest. In the winter season (November to January), when strawberries are most expensive in the Republic of Korea, early harvested fruit can provide greater price competition [3,45]. Therefore, the

W30 + N10 and W20 + N20 groups can support early harvests in order to increase the price competition of strawberries for farms and suppliers.

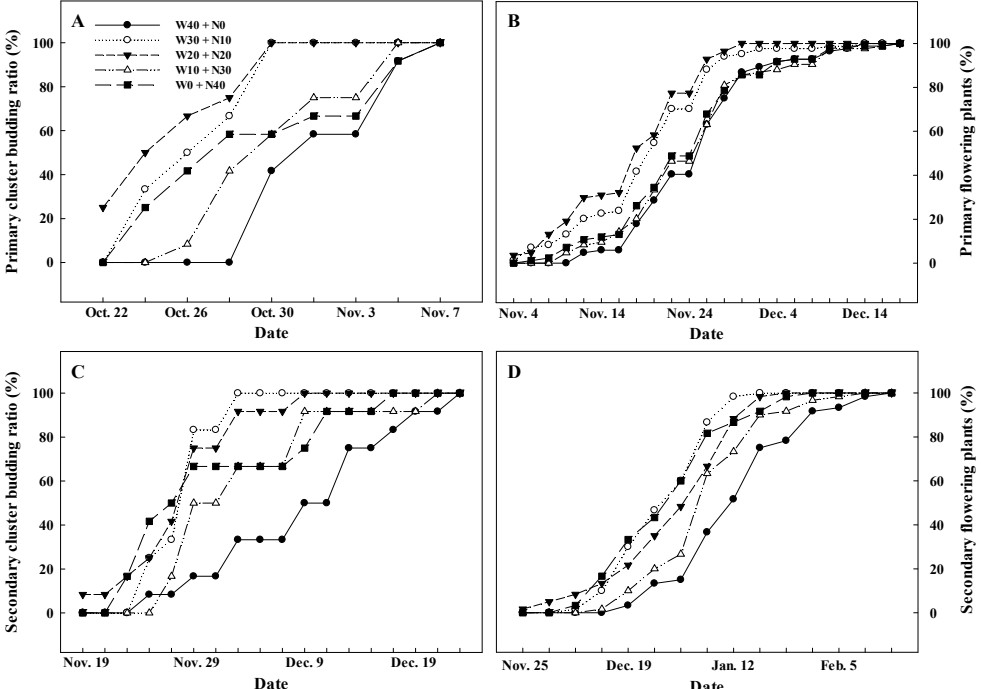

**Figure 3.** The primary cluster budding ratio (**A**), primary flowering plants (**B**), secondary cluster budding ratio (**C**), and secondary flowering plants (**D**) of "Maehyang" strawberries affected by the nutrient solution resupply period during the nursery stage. Refer to Figure 1 for details on nutrient solution resupply periods.

## 3.5. Fruit Characteristics and Quality

Table 5 summarizes the characteristics of fruits affected by the nutrient solution resupply period. Fruit firmness was significantly high in the W40 + N0 group, and acidity was lowest in the W0 + N40 group. It was reported that nitrogen can interfere with the accumulation of organic acid in fruit [46] and increase the production of nitrogen compounds, which extend the cell walls and make the fruit soft [47]. Accordingly, when changing the nitrogen concentration in the nutrient solution so as to increase the nitrogen concentration inside the strawberry plants, the fruit firmness and acidity decreased [48–50]. This observation is similar to a decrease in fruit firmness and acidity when nitrogen is increased in other fruits and vegetables, such as apples, cucumbers, kiwifruit, and tomatoes [51–54]. Fruit quality and yield are highly correlated with the EC level of the nutrient solution [55]. However, in this research, the nutrient resupply period was the only difference before the transplanting. After that, the fruits were cultivated in identical environments, with the same EC level of the nutrient solution at the same time until the harvest stage. Therefore, the fruit lengths, fruit diameters, fruit weights, and soluble solid contents of the fruits were not significantly different between all treatments.

**Table 5.** The fruit characteristics from the primary inflorescences of "Maehyang" strawberries affected by the nutrient solution resupply period during the nursery stage (*n* = 12).

| Treatment [z] | Fruit Length (mm) | Fruit Diameter (mm) | Fruit Weight (g/plant) | Fruit Firmness (N/$\Phi$5 mm) | Soluble Solids Content (Brix) | ACIDITY (%) |
|---|---|---|---|---|---|---|
| W40 + N0 | 51.6 a [y] | 29.0 a | 19.4 a | 4.08 a | 12.7 a | 1.01 ab |
| W30 + N10 | 50.4 a | 31.2 a | 18.6 a | 3.79 ab | 13.0 a | 0.95 ab |
| W20 + N20 | 49.7 a | 28.2 a | 18.0 a | 3.49 b | 13.2 a | 1.03 a |
| W10 + N30 | 48.6 a | 28.8 a | 18.1 a | 3.49 b | 12.3 a | 0.98 ab |
| W0 + N40 | 48.9 a | 27.7 a | 17.0 a | 3.53 ab | 12.6 a | 0.90 b |

[z] Refer to Figure 1 for details on nutrient solution resupply periods. [y] Mean separation within columns by Tukey's test at $p \leq 0.05$.

### 3.6. Fruit Yield of Strawberry

The fruit yield in December significantly decreased when the nutrient solution resupply period increased above W10 + N30 (Figure 4A). Excess nitrogen can inhibit flower formation and fruit yields [6,56]. There were no significant differences in the fruit yield in February and the total fruit yield. The marketable fruit yield was not significantly different between all treatments except in December (Figure 4B). This is similar to previous research that controlled the nutrient solution supply timing of strawberry runner plants [8,54]. The total fruit yield of strawberries shows a low level of interaction with the supply timing of the nutrient solution and is highly influenced by the concentration of the nutrient solution [57].

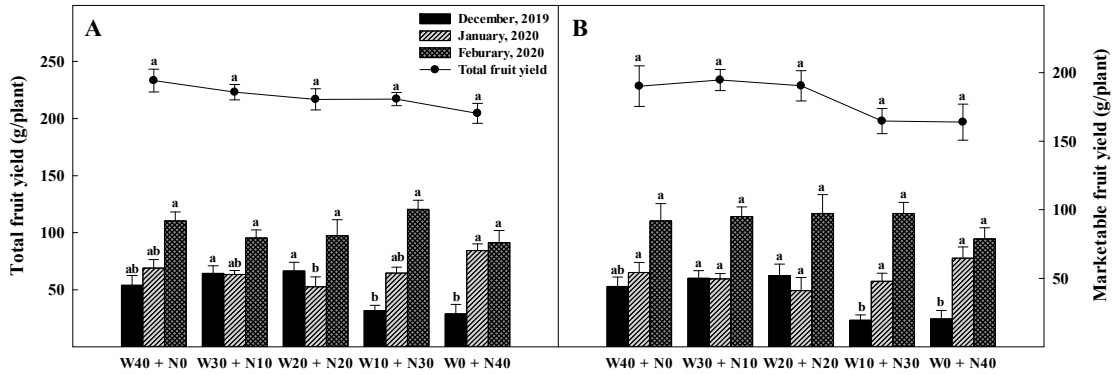

**Figure 4.** The total fruit yield (**A**) and marketable fruit yield (**B**) of "Maehyang" strawberries affected by the nutrient solution resupply period during the nursery stage. Refer to Figure 1 for details on nutrient solution resupply periods. Vertical bars indicate standard errors of the means (*n* = 12). Tukey's test at $p \leq 0.05$.

## 4. Conclusions

In this study, we aimed to determine the optimal nutrient solution resupply period for accelerating flower bud differentiation during a temporary nutrient-withholding period within the nursery period of "Maehyang" strawberries. The treatment of different nutrient solution resupply periods showed rapid flower bud development, a primary cluster budding ratio, and flowering plants without a difference in fruit or runner plant quality nor the total fruit yield. However, the early fruit yield showed a tendency to decrease when the nutrient solution resupply period increased above W10 + N30. Therefore, we suggest that the W20 + N20 treatment is the optimal nutrient solution resupply period for rapid flower bud differentiation without experiencing a decrease in fruit quality or yield.

**Supplementary Materials:** The following are available online at http://www.mdpi.com/2073-4395/10/8/1127/s1. Figure S1: Microscopic view of flower bud development stages on the apical meristem of "Maehyang" strawberry runner plants affected by the nutrient solution resupply period at transplanting day. The flower bud development stage was classified into 7 stages (0 to 6). The stages are as follows: 0, the vegetative apex stage; 1, the early apex enlargement stage; 2, the middle apex enlargement stage; 3, the later apex enlargement stage; 4, the apex division stage; 5, the sepal development stage; 6, the stamen development stage.

**Author Contributions:** Conceptualization, S.J.H.; methodology, S.J.H. and H.S.H.; formal analysis, H.S.H., H.W.J., H.R.L., H.G.J., and H.M.K.; resources, S.J.H.; data curation, H.S.H.; writing—original draft preparation, H.S.H.; writing—review and editing, S.J.H., H.W.J., and H.R.L.; project administration, S.J.H.; funding acquisition, S.J.H., H.S.H., H.W.J., H.R.L., and H.G.J. All authors have read and agreed to the published version of the manuscript.

**Funding:** This research was funded by the Technology Development Program for Agriculture and Forestry, Ministry for Food, Agriculture, Forestry and Fisheries, Republic of Korea (Project No. 315004-5).

**Conflicts of Interest:** The authors declare no conflict of interest.

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
