# Peer review of "Acceleration of Flower Bud Differentiation of Runner Plants in “Maehyang” Strawberries Using Nutrient Solution Resupply during the Nursery Period"

_agronomy, doi:10.3390/agronomy10081127_

Round 1

Reviewer 1 Report

Please, give more information about "runner plants" before planting: is not clear which kind of plant do you use on 9th August (tip runner? plug plant? dimension? number of leaf? Adding a photo may be good.

Row 94: during treatment period, "tap water only" was supplied 3 times a day, axactly as the intake of the nutrient solution? 

Row 94 and others: would be interesting to know the share of nutrient solution used by plants and the one not used and lost percolation. Has a calcolation been made of how many mineral elements were used by the plants?

Author Response

Reviewer 1

▪Please, give more information about "runner plants" before planting: is not clear which kind of plant do you use on 9th August (tip runner? plug plant? dimension? number of leaf? Adding a photo may be good.

→We added the ‘A strawberry (Fragaria ×ananassa Duch. cv. Maehyang) runner plants (the plug plant had 4 to 5 leaves, 9 to 10 mm of crown diameter, and transplanted on the same time which not started flower bud differentiation) were used for study.’ (line 85-87).

▪Row 94: during treatment period, "tap water only" was supplied 3 times a day, exactly as the intake of the nutrient solution?

→We add the ’The tap water was supplied at the same time with the nutrient solution.’ (line 97).

▪Row 94 and others: would be interesting to know the share of nutrient solution used by plants and the one not used and lost percolation. Has a calculation been made of how many mineral elements were used by the plants?

→ Unfortunately, we did not analyze the elemental content of plants before the treatment. And, we do not have any data of drain water. So, we cannot calculate about the used elements of plants. When planning a supplementary experiment, I will try to reflect on your advice and proceed with the experiment. Thank you very much for your good opinion.

  • The revisions are marked in blue.

Reviewer 2 Report

General Remarks:

  • English needs to be improved significantly. highly suggest to pay a scientific translation service to clean up grammar and sentence structures.
  • Using ‘tendencies’ to proof a hypothesis is not good scientific practice and should be avoided, and instead focus should be given to statistical significant data.
  • I would suggest to replace the term ‘runner plants’ with the correct term ‘daughter plants’.
  • Material and Methods: Study design is not clear (number of replicates? Number of plants involved in the study? Randomized block design?)

Abstract:

27-28: The treatments are not clear: What does 40, 0, 10, 20, and 30 stand for? mM/L, ppm, %, time? Needs to be clarified. Please also specify the Nutrient Solution used.

29 ff.: please only mention tendencies if they are statistically significant. If not significant, don’t mention them. If significant, please indicate clearly what are the differences.

32ff: How fast? Compared to what? Please indicate numbers

34ff: How fast? Compared to what?

Introduction:

42: Fragaria ×ananassa Duch. (use a multiply sign instead of ‘x’ and no space between multiply sign and ‘ananassa’.)

56-59: That’s very bad English. While the authors make a good point, the grammar and sentence structure make it almost impossible to follow.

71-73: Please state the hypothesis of the study. Here we hypothesize that xyz…

75-76: This sentences belongs into the last paragraph of the introduction

Material and Methods:

84-95:

  • How did you make sure that in- and out coming nutrient solutions were consistent in their nutrient content over the experiments time-period? Did you measure nutrient content of input and flow-through? What was the drainage rate over the time of the experiment?
  • Did you use a 1:1 nutrient solution and a tank system? Or did you use a concentrated nutrient solution and a 1:x rate (Dosatron)? If 1:1, how many stock solutions did you use? How did you make sure that he pH and EC was consistent every time you refilled the tank? If 1:x rate: how did you adjust the pH?
  • Did you water the coconut coir before transplanting? If yes, for how long?

95-108:

  • How did you control the EC gradually?
  • What were the average photoperiod and light intensity in the hydroponic system?
  • Please explain in one sentence how you have controlled flowering and fruit set?

109-122:

  • Please explain the experimental design: How many plants were planted? What was the experimental setup? Randomized block design? How many plants per replicate?

123-135:

  • Is that 12 daughter plants per treatment? How many daughter plants per mother plant and per replicate?
  • Did you count the number of developed daughter plants per treatment?

Results and Discussion:

183 – 211:

  • Were there difference in the number of flower buds per daughter plant?
  • It looks like W20+N20 and W10+N30 have the same result in terms of flower bud development by the day of transplanting? How is that explainable? And was the number of buds per plant and daughter plant per mother-plant recorded? For a nursery grower, it would be good to know if both treatments also performed similar in those categories.

212-233:

  • Please indicate what T-N is
  • Tendency? Please say whether or not it was significant

234-269:

236-237: It doesn’t seem that shoot length are growing across treatments. Dry weights are all the same. Please delete sentence.

270ff:

Please explain in M+M how preimary and secondary budding ratio is calculated (ratio of what to what?), how percent of primary and secondary flowering plants is calculated.

Table 5: Acidity is usally give in g/L tritratable acid.

317ff:

Tendencies are not significant and should not be discussed. Please delete sentence

Author Response

Reviewer 2 

General Remarks:

▪English needs to be improved significantly. highly suggest to pay a scientific translation service to clean up grammar and sentence structures.

→This article has already undergone English language editing by MDPI (Please check the certificate attached below) before submitting this article. But we agreed that some sentence can change in simple and easy to understand way. So, we changed the sentence which you pointed out. Thank you for your good advice.

▪Using ‘tendencies’ to proof a hypothesis is not good scientific practice and should be avoided, and instead focus should be given to statistical significant data.

→We changed the sentence in line 31, 218-220, 224-225, 242, 252, 252-254, and 321-322.

▪I would suggest to replace the term ‘runner plants’ with the correct term ‘daughter plants’.

→We did not use the mother plants in this experiment. We used the plug plants which already cut from mother plants. So, we used the term ‘runner plant’ to help readers understand that plant material is independent of the mother plant.

▪Material and Methods: Study design is not clear (number of replicates? Number of plants involved in the study? Randomized block design?)

→We changed the sentence ’The experimental treatments were laid out in a completely randomized block design. Each treatment included 72 plants. The 24 plants were used per replicate. And, replication was in 3 times.’ (line 177-178).

Abstract:

▪27-28: The treatments are not clear: What does 40, 0, 10, 20, and 30 stand for? mM/L, ppm, %, time? Needs to be clarified. Please also specify the Nutrient Solution used.

→We added the ‘; each number represents the days of treatment period). The nutrient solution treatments supplied using strawberry nutrient solution developed by Gyeongsangnam-do Agricultural Research and Extension.’ (line 28-30).

▪29 ff.: please only mention tendencies if they are statistically significant. If not significant, don’t mention them. If significant, please indicate clearly what are the differences.

→We deleted the sentence in line 31.

▪32ff: How fast? Compared to what? Please indicate numbers

→We added ‘The W20 + N20 group showed fast flower bud development stage at 10 days before transplanting and the day of transplanting (2.2 and 5.5), 6 days fast in a primary cluster budding ratio, and 16 days fast in flowering plants than the other groups.’ (line 33-36).

▪34ff: How fast? Compared to what?

→We add the ‘There was no difference in fruit characteristics by the treatments.’ (line 36).

Introduction:

▪42: Fragaria ×ananassa Duch. (use a multiply sign instead of ‘x’ and no space between multiply sign and ‘ananassa’.)

→We changed them all. Thank you for your advice.

▪56-59: That’s very bad English. While the authors make a good point, the grammar and sentence structure make it almost impossible to follow.

→We changed the sentence ‘Another way to promote the flower bud differentiation of strawberries is by changing the nitrogen levels inside the strawberries. When nitrogen levels increasing, carbohydrates are used for protein synthesis, and vegetative growth is stimulated’ (line 57-59).

▪71-73: Please state the hypothesis of the study. Here we hypothesize that xyz…

→We added the ‘Here we hypothesize that the optimal nutrient solution resupply after the temporary nutrient withholding period can promote the flower bud differentiation of strawberry.’ (line 73-75).

▪75-76: This sentences belongs into the last paragraph of the introduction

→We moved the sentence into line 81-82. Thank you.

Material and Methods:

84-95:

▪How did you make sure that in- and out coming nutrient solutions were consistent in their nutrient content over the experiments time-period? Did you measure nutrient content of input and flow-through? What was the drainage rate over the time of the experiment?

→We did not collect or measure the drain water. The strawberries were supplied the water from one same mixed tank. So, we sure about that the strawberries were supplied same water in same time with same amount and elemental contents.

▪Did you use a 1:1 nutrient solution and a tank system? Or did you use a concentrated nutrient solution and a 1:x rate (Dosatron)? If 1:1, how many stock solutions did you use? How did you make sure that he pH and EC was consistent every time you refilled the tank? If 1:x rate: how did you adjust the pH?

→We made 100 L of 100x A solution (12.00 kg of Ca(NO3)2·4H2O, 1.85 kg of KNO3, and 230.8 g of Fe-EDTA), and B solution (5.43 kg of KH2PO4, 4.32 kg of MgSO4·7H2O, 0.16 kg of K2SO4, 0.8 kg of NH4NO3, 29.4 g of H3BO3, 20.0 g of MnSO4·4H2O, 1.6 g of CuSO4·5H2O, 1.0 g of Na2MoO4·2H2O, and 8.7 g of ZnSO4·7H2O). And we put the same volume of A and B solution with fresh-water which already finish the element analysis into a mixed tank for nutrient solution. The pH was adjusted by adding nitric acid (HNO3) at 60% concentration. And, we add the ‘We controlled the EC and pH level using pH/EC meter (HI-98130, Hanna Instruments Co. Ltd., Woonsocket, RI, USA).’ (line 97-98).

▪Did you water the coconut coir before transplanting? If yes, for how long?

→We added the ‘which watered with tap water in 72 hours.’ (line 89).

95-108:

▪How did you control the EC gradually?

→We added the ‘(EC 0.6 dS·m-1 at early transplanting stage, EC 0.8 dS·m-1 at budding stage, EC 1.0 dS·m-1 at flowering stage, and EC 1.2 dS·m-1 at harvesting stage)’ (line 104-106).

▪What were the average photoperiod and light intensity in the hydroponic system?

→Regrettably, there was no recordable facility for photoperiod and light intensity inside the greenhouse. So, we could not measure the photoperiod and light intensity in the greenhouse.

▪Please explain in one sentence how you have controlled flowering and fruit set?

→We changed the sentence ‘We removed the flower to control the fruit set at 7 fruits on the primary inflorescence and 5 on the secondary inflorescence [5].’ (line 106-107).

109-122:

▪Please explain the experimental design: How many plants were planted? What was the experimental setup? Randomized block design? How many plants per replicate?

→We added the’ The experimental treatments were laid out in a completely randomized block design. Each treatment included 72 plants. The 24 plants were used per replicate. And, replication was in 3 times.’ (line 177-178).

123-135:

▪Is that 12 daughter plants per treatment? How many daughter plants per mother plant and per replicate?

→ We used the independent plug plant for the experiment, not the mother plants. Each mother plants, which for producing the plug plants, were cultivated in the same greenhouse and environment.

▪Did you count the number of developed daughter plants per treatment?

→We did not propagate the daughter plant from mother plants.

Results and Discussion:

183 – 211:

▪Were there difference in the number of flower buds per daughter plant?

→Only one flower bud can be made from the apical meristem. We checked how much flower buds developed fast by treatments. We checked inside of apical meristem with microscope.

▪It looks like W20+N20 and W10+N30 have the same result in terms of flower bud development by the day of transplanting? How is that explainable? And was the number of buds per plant and daughter plant per mother-plant recorded? For a nursery grower, it would be good to know if both treatments also performed similar in those categories.

→The nutrient resupply period can make fast flower bud differentiation. You can see the W30 + N10, W20 + N20, and W10 + N30 have more developed flower bud than W40 + N0 (none supplied nutrient solution) and W0 + N40 (continuous supplied nutrient solution). We think that W20 + N20 and W10 + N30 were appropriate nutrient resupply treatment for flower differentiation than W30 + N10. For nursery grower, the W20 + N20 was better treatment for saving fertilizer.

212-233:

▪Please indicate what T-N is

→T-N means total nitrogen. It already indicated in line 31-32 and 147 and Table 2.

▪Tendency? Please say whether or not it was significant

→We added the ‘When the nutrient solution supply period was increased, the T-N, P, K, and S concentrations of runner plants significantly increased. And the W0 + N40 showed the highest value than other treatments.’ (line 218-220), and ‘The C/N ratio showed a significant decrease when the nutrient solution resupply period was increased. And the W0 + N40 showed the lowest value than other treatments.’ (line 224-225).

▪234-269:

→We deleted sentence in line 242, 252. And changed the sentence ‘The fresh and dry weights of roots showed significant decrease when increasing the nutrient solution resupply period. The W0 + N40 was the lowest in fresh weights of roots. And, the W10 + N30 was the lowest in dry weights of roots.’ (line 252-254)

▪236-237: It doesn’t seem that shoot length are growing across treatments. Dry weights are all the same. Please delete sentence.

→We deleted the sentence in line 242. Thank you for your advice.

270ff:

▪Please explain in M+M how primary and secondary budding ratio is calculated (ratio of what to what?), how percent of primary and secondary flowering plants is calculated.

→We added the ’The number of first budding and flowering runner plants from primary and secondary inflorescences was investigated. The budding ratios and flowering ratios of each plant to total runner plants in the primary and secondary inflorescences were calculated.’ (line 161-163).

▪Table 5: Acidity is usually give in g/L tritratable acid.

→Our fruit acidity meter (GMK-835N, GMK Co. Ltd., Seoul, Republic of Korea) use ‘%’ for total acidity. This machine is designed to measure the total percentage of citric acid, tartaric acid and malic acid in sampled fruit juice from strawberries.

317ff:

▪Tendencies are not significant and should not be discussed. Please delete sentence

→The fruit yield from W10 + N30 and W0+ N40 was significantly low in December. We add the ’The fruit yield in December significantly decreased when the nutrient solution resupply period increased above W10 + N30 (Figure 4A).’ (line 321-322).

  • The revisions are marked in red.
